# Heritage Urbanism

## Mladen Obad Šćitaroci and Bojana Bojanić Obad Šćitaroci *

Faculty of Architecture, University of Zagreb, Zagreb 10000, Croatia; scitaroci@gmail.com
* Correspondence: bbojanic@arhitekt.hr; Tel.: +385-98-668-754

**Abstract:** Heritage urbanism considers the revitalization and enhancement of cultural heritage in spatial, urban, and landscape contexts, and it explores models for its inclusion in contemporary life. The main research question is whether it is possible, based on a number of case studies, to recognize models of the future use of heritage and interpret them as general models that may be applied to numerous specific cases. In doing so, the experience of the past becomes relevant and applicable to contemporary heritage revitalization and enhancement projects. The goal of the paper is to present Heritage Urbanism approach as an integral view of heritage in line with the ideas of sustainable development. Heritage is not viewed as isolated objects but rather as part of the immediate and wider environment. The context/environment affects heritage and its revival, while finding new uses and repurposing heritage has a stimulating effect on the environment and its development. The effects of this interaction can make heritage recognisable and can stimulate its sustainability. The survival and future of heritage are linked to urban and spatial planning, which takes into account the integrity of space and the cultural heritage in it. Urban and spatial planning methods are used. When these methods are enriched by the heritage urbanism approach, the result is the creation of specific methods that supplement well-known methods. In this context, cultural heritage can be used for place branding, infrastructure development, as a crucial element of urban design, or in other ways that aim to achieve an integral view of cultural heritage. The integral view requires the concerted action of different fields, such as regional development, the economy, tourism, transportation, and infrastructure. A fragmented and selective approach does not yield results.

**Keywords:** heritage urbanism; cultural heritage; heritage sustainability; heritage revitalization; heritage enhancement models

---

## 1. Introduction

*Heritage urbanism* is a term created and developed within a research project titled *Urban and Spatial Models for the Revival and Enhancement of Cultural Heritage* conducted at the Faculty of Architecture of the University of Zagreb. The project lasted five years, from 2014 to 2018, and was funded by the Croatian Science Foundation. Project participants consisted of 36 researchers from Croatia and a large number of foreign researchers who used the heritage urbanism approach in their research. The project encompassed 17 doctoral studies.

Heritage urbanism as a term and approach was first introduced to the public at an international conference under the title Cultural Heritage—Possibilities for Spatial and Economic Development held in Zagreb in 2015. The Proceedings of the Conference [1] containing 142 research papers by 227 authors from 39 universities were also published in 2015. In 2017, a second conference was held, followed by its Proceedings titled Models of Revitalization and Enhancement of Cultural Heritage—Multidisciplinary Dialogue [2], which further developed the concept of heritage urbanism. The approach and method of heritage urbanism were presented in two books published by Springer: *Quality of Life in Urban Landscapes* [3] and *Cultural Urban Heritage* [4].

Heritage urbanism considers the revitalization and enhancement of cultural heritage in spatial, urban, and landscape contexts, and it explores models for its inclusion in contemporary life. Heritage is not viewed as isolated objects but rather as part of the immediate and wider environment. The context/environment affects heritage and its revival, while finding new uses and repurposing heritage has a stimulating effect on the environment and its development. The effects of this interaction can make heritage recognizable and can stimulate its sustainability. The survival and future of heritage are linked to urban and spatial planning, which takes into account the integrity of space and the cultural heritage in it. The integral view of cultural heritage requires the concerted action of different fields such as regional development, the economy, tourism, transportation, and infrastructure.

### 1.1. Motivation

There are numerous examples of heritage in the world, often in a poor state, which require renewal and enhancement. Countless international declarations and charters relating to this matter have been adopted. Despite these, and irrespective of the laws protecting cultural heritage, heritage still decays, and not enough is invested in its restoration. The idea behind the heritage urbanism approach is to contribute to the enhancement of heritage and create motivation to start seeing heritage as an active subject in space, where its emanation is felt, instead of viewing it as a static object.

Over 25 centuries of urban culture in Croatia were the direct stimulus for this—from Illyrian proto-urban settlements (Illyrians, the contemporaries of the Etruscans and Celts) and the first Greek towns in Dalmatia (the eastern shore of the Adriatic Sea) to Roman towns and onwards until today. Preserved heritage shows the capacity of countless generations to inherit valuable heritage and build new structures within the old urban landscape.

### 1.2. The Importance of the Subject and of Cultural Heritage Enhancement

Issues concerning the preservation, revitalization, and enhancement of cultural heritage have been a subject of interest around the world for decades. This subject has prompted many international, national, and local institutions, experts, and scientists to contemplate the possibilities and importance of renewing and including heritage into the daily life of the community. In the context of the heritage urbanism approach, heritage is explored over a wide typological range and over a long period of time—from cultural landscapes to single buildings, from Greco-Roman archaeological sites to modernist architecture, from the tangible to intangible heritage. This includes using different perspectives—interdisciplinary, multidisciplinary, and transdisciplinary—to gain understanding of heritage through the interaction and overlapping of various expert and scientific approaches. This method is expected to bring new and different results, free of the usual patterns that often do not provide satisfactory results in terms of the revitalization and enhancement of cultural heritage.

### 1.3. Research Questions

The main research question is whether it is possible, based on a number of case studies, to recognize models of the future use of heritage and interpret them as general models that may be applied to numerous specific cases. In doing so, the experience of the past becomes relevant and applicable to contemporary heritage revitalization and enhancement projects.

Authentic heritage is a nonrenewable resource because any replica of it is not the original, regardless of how faithful to the original it may be. Countless examples of decaying, dying, and inactive cultural heritage builds have stimulated a re-examining of usual methods and an exploration of potential new approaches and effective solutions to give new life to cultural heritage. This new life should spur the creation of new spatial and social contexts where heritage becomes a success, a resource that is no longer a burden on the community. For heritage to be the driving force of its activities, its renewal and revitalization must respond to contemporary needs (i.e., it must be capable of adapting to contemporary requirements of communities that inherit and maintain it). Thus, the main research questions aim to determine the criteria for new interventions in heritage and the models of

heritage use in order to include heritage in the life of towns and settlements. Research initially sought answers in the context of urban and spatial planning, then it broadened its scope by including different perspectives such as multidisciplinary views and the objectification of heritage revitalization.

*1.4. Literature*

Plenty of papers on cultural heritage have been written and published, particularly those from the cultural and historic perspective [5–7]. Not as many papers offer a conservationist view [8–10], and only a small number deal with the aspect of methodology [11]. There is little literature on the methodological aspect of cultural heritage in the contexts of architecture, urbanism, and spatial planning [12]. Literature and research on activating passive heritage, the long-term survival of heritage, and the criteria and models of revitalization and enhancement of heritage are limited. Therefore, these topics serve as initial research questions in the development of the heritage urbanism approach. This also underlines the importance of newly published research papers, which might contribute to the enhancement of cultural heritage by offering a different, specific approach. Numerous papers have been published on cultural heritage management [13–15] as well as on the subject of adaptive reuse [16]. An important starting point of the heritage urbanism approach was the Recommendation on the Historic Urban Landscape (HUL), 2011 [17]. This Recommendation focused mainly on application in the local context.

*1.5. Understanding of Cultural Heritage*

We tend to perceive cultural heritage as a relic of the past that must be protected and preserved. It is often in a poor state, it is static, and is without a long-term purpose. Instead of incorporating it in contemporary life, it is viewed as a problem. We frequently see cultural heritage as isolated built artefacts, forgetting that it was made and has been neglected by people, and that without people and life in it, there is no revitalization or sustainability. As people's habits, ideologies, and cultural landscapes change, the life of heritage also changes.

The heritage urbanism approach considers cultural heritage in the broadest sense—from single buildings, settlements, and towns to cultural landscapes. Research includes a large number of examples of different types of heritage: designed and associative cultural landscapes, archaeological heritage, fortification architecture, manors and countryside culture, rural heritage, island and coastal heritage, the historic urban landscape, small towns, modernist architecture and 20th century urbanism, garden heritage, place branding, public areas connecting the city, soundscape, natural light as heritage, and so on.

The heritage urbanism approach is presented under the next three titles. This approach aims to enhance cultural heritage, given that it is endangered and is disappearing because of the lack or change of use, or for other reasons. The heritage urbanism approach will be compared to other approaches, both other views on urbanism and approaches focusing on cultural heritage.

## 2. The Heritage Urbanism Approach

In the Strategy of Conservation, Protection and Sustainable Economic Development of Cultural Heritage of the Republic of Croatia for the Period 2011–2015, it is stated, among other things, that there is a methodology for creating documentation needed for the renewal of cultural heritage (which causes excessively slow and inappropriate renewal) and no clear criteria for the evaluation of heritage. This was a stimulus for research focusing on the context of urbanism, which views not only the historic building, but also the area that surrounds it. Considering over 25 centuries of urban culture on Croatian soil (from the first Greek towns dating back to 5th century BC), the goal was to develop an approach that could offer options for creating an acceptable and sustainable future of heritage, which is very much needed and evidenced by the poor state of a number of cultural heritage sites of all types and levels—from individual buildings to historic towns and cultural landscapes, and from local sites to

world heritage sites protected by UNESCO. Two key words—cultural heritage and urbanism—led to the creation of the term HERITAGE URBANISM.

Heritage urbanism considers the revitalization and enhancement of cultural heritage in spatial, urban, and landscape contexts, and it explores models for its inclusion in contemporary life. In search of revitalization and enhancement models, heritage is not viewed as isolated buildings/objects, but rather as part of the immediate and wider environment. The context/environment affects heritage and its revival, while finding new uses and repurposing heritage has a stimulating effect on the environment and its development. The effects of this interaction can make heritage recognisable and can stimulate its sustainability.

The approach uses an integrated/integral view of heritage (integral preservation) in line with the ideas of sustainable development. The survival and future of heritage are linked to urban and spatial planning, which takes into account the integrity of space and the cultural heritage in it. Thus, heritage is viewed as part of a wider living space and of the local community—as a settlement/town or a cultural landscape. Urban and spatial planning methods are used. When these methods are enriched by the heritage urbanism approach, the result is the creation of specific methods that supplement well-known methods. In this context, cultural heritage can be used for place branding, infrastructure development, as a crucial element of urban design, or in other ways that aim to achieve an integral view of cultural heritage.

The integral view of cultural heritage requires the concerted action of different fields, such as spatial planning, regional development, the economy, tourism, fiscal policy, transportation and infrastructure, and so on. A fragmented and selective approach does not yield results, as confirmed by examples of cultural heritage renewal, when the renovation is only done according to conservation criteria without taking into account economic criteria and long-term sustainability.

The heritage urbanism approach is to be used when the buildings and spaces of cultural heritage have no long-term purpose or no purpose at all. When heritage is not used, it decays very quickly and gradually disappears, so it is necessary to find appropriate models for its activation. The effects of active protection that allows for the continuity of life and people's activities have been explored since the middle of the 20th century. The active use of heritage is a guarantee for its survival, which is why it is said that heritage does not need to be protected from people, but for people. We need to learn to live with heritage, but also from heritage.

Cultural heritage needs to be managed, cared for, and enhanced. In the 19th century, John Ruskin emphasised the importance of maintaining cultural heritage to minimize the need for its renewal [18]. When we stop taking care of heritage, it very quickly decays and disappears. Today, people want to make profit from heritage (cultural tourism), but they do not want to invest into enhancing it. This increases the importance of heritage management [19].

From the perspective of the economy, the combination of tourism and cultural heritage is seen as appropriate and successful. Tourism lives off heritage by using beautiful landscapes, the picturesqueness of the environment, and inherited cultural goods. The tourism industry often misuses and impoverishes heritage, investing too little or nothing at all in heritage rehabilitation [20,21]. This is becoming a global issue visible at all levels. The consequences of such behaviour have been evident for many years in the cultural heritage protected by UNESCO (Venice, Dubrovnik, etc.).

Based on the above, heritage urbanism sets three methodological levels aiming to recognize/determine/define:

1. identity factors, factors of effect, and value factors;
2. evaluation criteria, enhancement criteria, and criteria for new interventions; and
3. cultural heritage revival and enhancement models.

The heritage urbanism approach is tested and developed through doctoral research dealing with specific topics and cases. This process continues, even after the research has ended, extending to new doctoral research. Practical verification of the approach in several renewal and revitalization projects

involving manor houses and their surroundings is underway. Further application of the approach and its verification have just begun, and this requires some time, so it is not yet possible to speak of the results of practical verification.

Evaluation criteria are determined by recognising the factors of identity (identity features). The factors of effect indicate external influences that affect heritage and change it. The factors of identity are determined by experts, preferably from different fields and different viewpoints. Identification of these factors is based on historical analyses of various cartographic and written sources, the current situation on the site, and data gathered from literature. The evaluation is made by the experts, but it is also recommended to include the community through questionnaires, workshops, and roundtables. The value factors (value features) of heritage are a starting point for determining criteria for enhancement and new interventions. Criteria for new interventions guide the selection or creation of models of heritage revival and enhancement. Whichever model or models are used, they must stimulate new, high-quality interventions that enhance and enrich heritage instead of destroying it. One day, these new interventions will perhaps be recognized as new heritage.

The final goal and result of the heritage urbanism approach is recognising and/or creating cultural heritage revival and enhancement models, which emerge from researching the historical and current/contemporary models. It is essential that the new models respect the heritage, ensure contemporaneity, and stimulate sustainable development. Seventeen models were recognized and divided into three groups: universal heritage models, basic heritage models, and thematic models of the heritage approach [4] (pp. 457–475).

Universal heritage models are used in all cases of heritage revitalization. They begin with heritage protection, include the urban and architectural concept, determine the way heritage is used, and contribute to legal certainty and economic sustainability.

Basic heritage models are based on considering heritage and its context from spatial and urban planning points of view, and they are related to the transformation of built heritage and linking heritage to sustainable development.

Thematic models stem from multidisciplinary and transdisciplinary views (architectural, aesthetic, ecological, tourist, experiential, etc.), which may offer interesting contemporary programs, ideas, and interpretations for the enhancement and revitalization of heritage.

Through the criteria of heritage management, models are categorized into protection models (preservation, maintenance, and renewal), development models (focusing on research, revitalization, adaptation, etc.), and models of use (relating to culture, education, science, technology, tourism, etc.).

Cultural heritage revitalization and enhancement models are meant to be used for the protection and renewal of cultural heritage, in urban planning, architectural and landscape design, and while managing cultural heritage and deciding on conservation. These models are not the solution, but they do provide a framework for finding a solution. Model selection depends on the context, the identity characteristics of the heritage, factors of influence, the level of evaluation, and the criteria for new interventions in the heritage.

Most often, several different but compatible basic and thematic models are used. They complement one another, creating a quality model mesh. Among a multitude of models and submodels, it is necessary to recognize and select appropriate models to enhance heritage, preserve its identity features, contribute to the active and sustainable use of heritage, and to assist in finding its long-term use to give heritage new life. The long-term use of heritage is a guarantee for lasting sustainability.

## 3. Heritage Urbanism and Other Perspectives

The historic urban landscape (HUL) approach is holistic and interdisciplinary. It focuses on historic cities. "It is based on the recognition and identification of a layering and interconnection of natural and cultural, tangible and intangible, international and local values present in any city. It includes social and cultural practices and values, economic processes, and the intangible dimensions of heritage as related to diversity and identity" (UNESCO, 2011) [17]. Although the starting points

are similar, heritage urbanism primarily focuses on potential models for revitalization and for the enhancement of different types of built heritage, not only historic cities.

If we compare heritage urbanism to other urbanist perspectives, we will see that a comparison is possible with about seventy (or more) different interpretations of urbanism focusing on a single subject. The urbanism approach implies a relationship between built structures and the space/context where this structure is located with the aim of achieving space that is optimal for living. We explore urban constants such as the urban landscape (built structures), vehicle and pedestrian flows, public areas and the landscape, infrastructural systems, etc. These urban constants are accompanied by urban parameters characteristic of various urbanist views: archaeological urbanism, blu urbanism, contest urbanism, ecological urbanism, environmental urbanism, green urbanism, infrastructural urbanism, landscape urbanism, new urbanism, postmodern urbanism, sustainable urbanism, traditional urbanism, and many others. Heritage urbanism is one of about seventy urbanist perspectives that focus on cultural heritage as seen through the prism of urban development and planning. This represents an interdisciplinary and multidisciplinary view on heritage in the context of the space that surrounds and influences it. A heritage revitalization program can enhance this space significantly [10].

An interdisciplinary approach is characteristic of urbanism. It includes urbanist, spatial, landscape, architectural, cultural and historical, conservation, technical and infrastructural, legal, economic, and other views on space.

The urbanist view studies heritage in the context of the city/settlement through an analysis of how cultural heritage builds the city and how it can contribute to the development of the city/settlement. Its goal is to include heritage into the life of urban and rural areas. This is an ongoing topic of contemporary discussion. For example, the 2030 Agenda for Sustainable Development (in particular target 11.4 of goal 11) [22] and the New Urban Agenda, both by the United Nations, make reference to the role of cultural heritage in the sustainable development of cities.

The spatial view considers heritage in the context of the region and landscape (i.e., on a large scale). This is particularly relevant for cultural landscape heritage and its inclusion in the life of the region and in the development of the local community [23].

The landscape view delves into the preservation and enhancement potential of valuable elements of the landscape, ambiance and visual experience of the city/settlement, or the cultural landscape where the heritage is located [24].

The architectural view focuses on attaining excellence in new/contemporary construction within heritage spaces. Modern architecture must respect the space and older buildings [25,26].

Cultural and historical views are used to explore the historical context and the genesis of space by examining the factors influencing the creation and changing of space in the past. The identity features of space/heritage recognized in this process serve as a starting point for new interventions in heritage [27].

Through the conservation view, heritage is seen as an artefact of the past. This view advocates renewal that affirms the relevant features and character of heritage while keeping its authenticity [9].

The legal view deals with questions of ownership and related issues that can facilitate or impede (i.e., speed up or slow down) heritage renewal procedures [19].

The economic view considers different scenarios of financial sustainability. This not only includes sustainability of the current renewal, but also sustainability of income and heritage use after its renewal, ensuring realistic and long-term funding sources [28,29].

The task of the ecological view is to preserve natural resources and the authenticity of the ambiance [30].

Each of these views is narrow and does not approach the issues of heritage holistically. The holistic approach is a crucial part of the heritage urbanism approach: considering different perspectives, building on them, and connecting them to enhance the cultural heritage. Heritage urbanism provides a long-term development plan, encourages active use of heritage, and aims to enhance the experience of authenticity. This holistic approach helps to objectify heritage revitalization in order to avoid making

subjective decisions on renewal and enhancement procedures for each individual case, which is usually a consequence of a lack of clear heritage evaluation criteria and criteria for new interventions, while cultural heritage management models are still in the making.

## 4. Urbanscape Emanation

The urbanscape emanation concept developed within the heritage urbanism approach is used to discover/recognize the "hidden", latent layers in space and the way they can be used and combined in the planning and further development of heritage and settlements/space. This concept stimulates the development of tools and methods within the urban development and planning of settlements, towns, and landscapes.

"Urbanscape emanation exposes space impressions, modifies insights, and examines the addition of time and structure to space. The transition from static models to dynamic models is what we wish to achieve by creating awareness about urbanscape emanation. The concept of emanation is seen as the impact of the unit, system, and/or being on its environment and what allows it to move forward in space or time. The design mode of a city is the choreography of motion, visual illusions, and soundscape anticipation. The aim is to create a paradigm independent of location, content, scale, time, and technology. It presents a network of key terms and concepts, considering the location, context, and programme, and integrating classification, structure, and analysis" [31].

Urbanscape emanation is a concept that detects multilayered values exerting influence on changes in urban space. The concept seeks to achieve urbanscape resilience under the pressure of urban development. It wishes to show the possibilities for space development by identifying parts of a city that can take on the most diverse city functions and that have the ability to support new, additional functions. Research raises the level of awareness and responsibility of actors and agencies in the urban landscape for the legacy of landscapes and its potentials through interacting thematic layers: urbanscape, naturescape, heritagescape, walkscape, soundscape, mindscape, ambiencescape, touristscape, waterscape, archaeologicalscape, publicscape, etc.

The aim is to explore and contribute to the community perception of placemaking and to adapt experience to all transformations in its immediate surroundings. The influences of heritage, sound, movement, experiences, and attractions, which overlap with the basic physical structure of the city system, will create an integrated and desirable living space. Planning this way boosts the power of urbanscape emanation. The result includes and adds hidden, forgotten, and new layers to public life.

Urbanscape, as with culture, is a process and is expressed in multitudinous forms. We can perceive the urbanscape by focusing on tension between the temporary and the permanent, between the planned and the experiential [32]. Any form of urbanscape is by definition a part of a shared story, a passion, a reflection of everyday life within a culture. This is why awareness of the urbanscape is a living phenomenon.

Urbanscape emanation considers historical, social, cultural, economic, technical, and urban research as equal starting points for interdisciplinary-based spatial planning. This type of research needs cooperation between universities and spatial planning practice to enable easier communication with different stakeholders, experiments in participatory approaches, and the constant search for areas to improve both education and planning scenarios and strategies. A multilayered overlap of a complex network of both people and data is made possible through contemporary technology and GIS-based tools. This overlap reveals both measurable features as well as all invisible, hidden, forgotten, and disappeared layers such as social and cultural constructs with psychological influences and with the intentions of raising awareness for the locals and investors. Awareness allows for future towns to draw memory in their landscapes as an added feature and value that can redefine private and public relations leading to better and stronger partnerships.

## 5. Conclusions

Today, experts are searching for ways to change the view of heritage from static to active and to make it a creative process to guarantee the sustainable and long-term use of cultural heritage. Based on this philosophy, many European cities have been able to develop their economy and tourism because they have invested in renewal, revitalization, and reuse of cultural heritage. Reusing historic buildings only as museums or cultural institutions is not sustainable. A change of perspective is desired because cultural heritage must be viewed holistically, in a multidisciplinary manner, from different perspectives, and by taking into account long-term sustainability criteria. This sort of approach can contribute to a more objective and realistic revitalization and quicker enhancement of cultural heritage. This way, instead of dead capital, which is often the case, heritage becomes capital for development. In the past, renewal focused on physical renewal and conservation. Today, besides the latter, the aim is also to interpret heritage and, thus, make it more visible. Cultural heritage management that encourages activities has also been introduced. It is expected to adapt heritage to the contemporary requirements of the community by following the principles of sustainability in order to include heritage in the daily life of towns and settlements. Only heritage that is active and in use is guaranteed to survive and achieve sustainability. Cultural heritage cannot survive if it is only formally protected. The main question is not how to prevent the decay of cultural heritage and preserve it. The question is what we can lose if cultural heritage is not included in spaces where people live and in the community to which the heritage belongs. Leaving old heritage to decay while constructing a new one is irrational.

Heritage urbanism is a dynamic approach to spatial development and enchantment, suitable for amendments and enhancements over time. Urbanscape emanation is a concept of the detection of multi-layered values and use in planning. Interconnected, they become a new process of a planning balance between multiple layers of urban landscape, heritage, and cultural tourism.

This special issue of *Sustainability* dedicated to the sustainability of culture and heritage is a significant contribution to the enhancement of heritage with new ideas and new possibilities. We would like to thank the editorial staff and colleagues Tigran Hass and Krister Olsson for their suggestion to dedicate this special issue to heritage urbanism.

**Author Contributions:** Conceptualization, M.O.Š. and B.B.O.Š.; Methodology, M.O.Š. and B.B.O.Š.; Validation, M.O.Š. and B.B.O.Š.; Investigation, M.O.Š. and B.B.O.Š.; Resources, M.O.Š. and B.B.O.Š.; Writing-Original Draft Preparation, M.O.Š. and B.B.O.Š.; Writing-Review & Editing, M.O.Š. and B.B.O.Š.

**Funding:** This research was funded by the Croatian Science Foundation grant number [2032].

**Conflicts of Interest:** The authors declare no conflict of interest.

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
