# Peer review of "Heritage Urbanism"

_sustainability, doi:10.3390/su11092669_

Reviewer 1 Report

1) Based on an “opinion”-type paper the authors claim the introduction of a new term/concept “Heritage Urbanism” that may enrich the approach to the “Sustainability of Culture and Heritage” by the creation of specific methods that supplement the well-known Urban and spatial planning methods.

2) Still according to the authors this new term and approach (“Heritage Urbanism”) was created and developed within the context of an international research project lead by the authors themselves.

3) Introduced for the first time to the public in 2015 at the international conference entitled Cultural Heritage - Possibilities for Spatial and Economic Development”, the concept of “Heritage Urbanism”, as a term and approach, was further developed and consolidated with the organization of a second international conference in 2017 and the publication of two books by Springer.

 4) Although the present paper seems to be supported on the experience gained by the authors themselves in an international context (this is through an international project, two international conferences and two books published by Springer), almost only their own work is cited in this paper as references.

5) So both the scientific and the academic values of the paper may eventually be improved either by the inclusion of a very singular, paradigmatic case study that illustrates, demonstrates and validates the application of this new term and approach or by the citation as references of works performed by others than the authors themselves.

Author Response

1.1. Although the present paper seems to be supported on the experience gained by the authors themselves in an international context (this is through an international project, two international conferences and two books published by Springer), almost only their own work is cited in this paper as references.

ANSWERS TO REVIEWER: In the supplemented text version we added the literature. Earlier, we only mentioned our work on heritage urbanism to show the way of developing ideas and a more detailed view of heritage urbanism. Now a lot of references are mentioned:

References

1. Obad Šćitaroci, M., Ed. Cultural Heritage - Possibilities for Spatial and Economic Development / Kulturno naslijeđe - prostorne i razvojne mogućnosti kulturnog naslijeđa, Faculty of Architecture University of Zagreb and Croatian Academy of Sciences and Arts: Zagreb, Croatia, 2015; ISBN 978-953-8042-10-2, ISBN 978-953-8042-11-9 (eBook).

2. Obad Šćitaroci, M., Ed. Modeli revitalizacije i unaprjeđenja kulturnog naslijeđa - multidisciplinarni dijalog, Faculty of Architecture University of Zagreb and Croatian Academy of Sciences and Arts: Zagreb, Croatia, 2017; ISBN 978-953-8042-29-4, ISBN 978-953-8042-30-0 (eBook). 

3. Obad Šćitaroci, M. Heritage as an Active Space and Spatial Resource. In Quality of Life in Urban Landscapes - In Search of a Decision Support System, Cocci Grifoni, R., D’Onofrio, R., Sargolini, R., Eds.; Springer: Cham, Switzerland, 2018, pp. 341– 348, ISBN 978-3-319-65580-2, ISBN 978-3-319-65581-9 (eBook), https://doi.org/10.1007/978-3-319-65581-9

4. Obad Šćitaroci, M.; Bojanić Obad Šćitaroci, B.; Mrđa A., Eds. Cultural Urban Heritage – Development, Learning and Landscape Strategies, Springer: Cham, Switzerland, 2019, ISBN 978-3-030-10611-9, ISBN 978-3-030-10612-6 (eBook), https://doi.org/10.1007/978-3-030-10612-6

5. European Cultural Heritage Strategy for 21th Century, Council of Europe, 2017, Available online:

https://rm.coe.int/16806f6a03 (accessed on 26 April 2019).

6. Smith, L. Uses of Heritage, Routledge (Taylor&Francis Group): London, and New York, 2006, ISBN10:0-415-31830-0, ISBN10:0-203-60226-9 (eBook).

7. Ashworth, G.J.; Graham, B.; Tunbridge, J.E. Pluralising past: heritage, identity and place in multicultural societies, London: Pluto Press: London, 2007.

8. Ashurst, J., Ed. Conservation of Ruins, Elsevier: London, 2006,

ISBN-13: 978-0750664295, ISBN-10: 0750664290.

9. Mendes Zancheti, S.; Simila, K., Measuring Heritage Conservation Performance, VI. International Seminar on Urban Conservation, CECI & ICCROM: Olinda, Brazil; Rome, Italy, 2012, CECI ISBN: 978-85-98747-16-3 ICCROM ISBN: 978-92-9077-230-9.

10. Cohen, N. Urban Planning Conservation and Preservation, McGraw-Hill: New York, 2001,

ISBN-13: 978-0071375849, ISBN-10: 0071375848.

11. Veldpaus,L.; Pereira Roders, A.R.; Colenbrander, B.J.F. Urban Heritage: Putting the Past into the Future, The Historic Environment, 2013, 4(1), 3–18.

12. Janssen, J.; Luiten, E.; Stegmeijer, E. Heritage as sector, factor and vector conceptualizing the shifting relationship between heritage management and spatial planning, European Planning Studies, 2017, 25(9), 1654-1672.

13. McManamon, F.P.; Hatton, A. Eds. Cultural Resource Management in Contemporary Society, Routledge (Taylor&Francis Group): London, 2000, ISBN 978-0-415-64241-5 (ISBN-13: 978-0415117852, ISBN-10: 0415117852).

14. McManamon, F.P. Ed. New Perspectives in Cultural Resource Management, Routledge (Taylor&Francis Group): London, 2017, ISBN 9781138101128.

15. Tunbridge, J.E.; Ashworth, G.J. Dissonant Heritage - the Management of the Past as a Resource in Conflict, John Wiley & Sons: Chichester, UK, 1996, ISBN-13: 978-0471948872, ISBN-10: 047194887X.

16. Baum, M.; Christiaanse, K. Eds. City as Loft - Adaptive Reuse as a Resource for Sustainable Urban Development, GTA Verlag (ETH Zurich): Zurich, Switzerland, 2012, ISBN 978-3-85676-302-2.

17. Recommendation on the Historic Urban Landscape (HUL), UNESCO: Paris, 2011, Available online: https://whc.unesco.org/en/hul/, https://whc.unesco.org/en/news/1026/, https://whc.unesco.org/uploads/activities/documents/activity-638-98.pdf (accessed on 26 April 2019).

18. Ruskin, J. Seven Lamps of Architecture, 8th ed.; Farar, Straus and Giroux: New York, 1981, ISBN: 978-0486261454.

19. Messenger, P.M.; Smith, G.S., Eds. Cultural Heritage Management: A global perspective, University Press of Florida, SAD, 2015, ISBN 978-0-8130-6085-9, ISBN-13: 978-0-8130-34607, DOI: 10.5744/florida/9780813034607.001.0001.

20. Leask A.; Fyall A. Managing world heritage sites. Routledge (Taylor&Francis Group): London and New York, 2006, ISBN-13: 978-0-7506-6546-9, ISBN-10: 0-7506-6546-7. 

21. McKercher, B.; du Cros, H. Cultural Tourism. The Partnership Between Tourism and Cultural Heritage Management, Haworth Hospitality Press (Routledge): London, 2002, ISBN 0‐7890‐1106‐9.

22. Transforming our World: the 2030 Agenda for Sustainable Development

A/RES/70/1, United Nations, Available online: https://sustainabledevelopment.un.org/ (accessed on 26 April 2019).

23. Eppich, R., Ed. Cultural Heritage Landscape and Rural Development, Good Practice, Methodology, Policy Recommendations & Guidelines for Rural Communities, 2014, ISBN: 978-84-697-1389-1.

24. Cultural Landscapes and Herirtage Values, University of Massachusetts Amherst: Amherst, SAD, 2015, Available online: https://www.umass.edu/chs/news/CLHVFinalProgram.pdf (accessed on 26 April 2019).

25. Lardinois, S.; Arato Gonçalves, A. P.; Matarese, L.; Macdonald, S., Eds. Contemporary Architecture in the Historic Environment - An Annotated Bibliography, The Getty Consertvation Institute: Los Angeles, SAD, 2015, Available online:  https://www.getty.edu/conservation/publications_resources/pdf_publications/pdf/cahe_bibliography.pdf (accessed on 26 April 2019).

26. Building in Context - New development in historic areas, English Heritage: London, CABE: London, 2001, Available online: https://www.designcouncil.org.uk/sites/default/files/asset/document/building-in-context-new-development-in-historic-areas.pdf (accessed on 26 April 2019).

27. Butina Watson, G.; Bentley, I. Identity by Design, Butterworth-Heinemann: Oxford, UK, 2007, ISBN-13: 978-0750647670, ISBN-10: 0750647671.

28. Licciardi, G.; Amirtahmasebi, R., Eds. The Economics of Uniqueness – investing in Historic City Cores and Cultural Heritage Assets for Sustainable Development, The World Bank Washington, D.S., 2012, ISBN (paper): 978-0-8213-9650-6ISBN (electronic): 978-0-8213-9706-0, DOI: 10.1596/978-0-8213-9650-6.

29. Navrud, S.; Ready, R.C., Eds. Valuing Cultural Heritage: Applying Environmental Valuation Techniques, Edward Elgar: Cheltenham, UK, 2002, ISBN: 1-84064-079-0.

30. Almo, F. The cultural landscape as a model for the integration of ecology and economiscs, Bioscience, 2000, 50(4), 313-320.

31. Bojanić Obad Šćitaroci, B. Urbanscape Emanation vs. Types of Landscape. In Quality of Life in Urban Landscapes; Cocci Grifoni, R., D’Onofrio, R., Sargolini, R., Eds.; Springer: Cham, Switzerland, 2018, pp. 349–355, https://doi.org/10.1007/978-3-319-65581-9

32. Bojanić Obad Šćitaroci, B. Perceiving Heritage vs. Awareness of Heritage. In Cultural Heritage – Possibilities for Spatial and Economic Development; Mladen Obad Šćitaroci, Ed. Faculty of Architecture University of Zagreb and Croatian Academy of Sciences and Arts: Zagreb, Croatia, 2015, pp. 212-215, ISBN 978-953-8042-10-2, ISBN 978-953-8042-11-9 (eBook).

1.2. So both the scientific and the academic values of the paper may eventually be improved either by the inclusion of a very singular, paradigmatic case study that illustrates, demonstrates and validates the application of this new term and approach or by the citation as references of works performed by others than the authors themselves.

ANSWERS TO THE REVIEWERS: The supplemented text is lines: 183-188

The heritage urbanism approach is tested and developed through doctoral research dealing with specific topics and cases. This process continues even after the research has ended, extending to new doctoral research. Practical verification of the approach in several renewal and revitalisation projects involving manor houses and their surroundings is underway. Further application of the approach and its verification have just begun, and this requires some time, so it is not yet possible to speak of the results of practical verification.

For illustrative purposes we list five accepted titles and synopsis of doctoral researches where the heritage urbanism approach is applied. Research works are:

- Integrated Protection Models of Archaeological Heritage in Dubrovnik's historic Area (Zehra Laznibat, Faculty of Architecture University of Zagreb)

- Zagreb Lower Town - Urbanistic Traits of the Eastern Part 1905-2010 (Marijana Sironic, Faculty of Architecture University of Zagreb)

- Urban Traits and Urbanity layer of the King Zvonimir Street in Zagreb (Dario Sironic, Faculty of Architecture University of Zagreb)

- Integrated spatial Protection and Management Models of rural Heritage in protected Areas  - Lonjsko polje Nature Park (Ksenija Petrić, Faculty of Architecture University of Zagreb)

- A Method for Contemporary Architectural Restitution of Croatian Country House Complexes  (Boris Dundović, Vienna University of Technology (TU Wien), Faculty for Architecture and Planning, Institute of History of Art, Building Archaeology and Restoration)

Reviewer 2 Report

The paper is interesting and deals with a very current topic in the international debate about the  relationship between cultural heritage and development and planning of cities.

The abstract should be integrated in order to clarify the goal of the paper that is not so clear since its beginning.

The paper is completely lacking in references about the discussed issue although it is a much debated topic at international level. The whole first paragraph should be integrated including bibliographic references (for example in lines 79-80; 121; 129-130; 199).

At the end of the first paragraph a brief summary about the structure of the paper and the content of the following paragraphs should be included in order to prepare readers for the structure of the paper and contribute to make the goal of the work clearer.

One of the latest documents about the holistic approach necessary in dealing with issues related to cultural heritage is the Historic Urban Landscape Recommendations by UNESCO (2011). This document is never mentioned in the paper. What is the relationship between Heritage Urbanism and UNESCO HUL approach? A section about this relationship should be included, considering the importance and the international interest that the UNESCO approach plays today.

In the paragraph 1.3 there are the research questions. The main research question dealt in this paper should be clearer underlined.

Lines 140-144: Are there some case studies in which this approach has already been implemented?

Lines 146-148: Who are the subjects involved in the identification of the value factors? How are they identified? Are the values identified by experts or also by the support of the community?

Lines 152-157: Which are the 17 recognized models? They should be explicitly cited in the paper or bibliographic references need to be included in order to help readers to identify them.

Lines 199-200: The contribution of cultural heritage to the development of cities is a great challenge today. In fact, some international debates refer to this topic. For example, the 2030 Agenda (in particular the target 11.4 of the goal 11) and the New Urban Agenda by United Nations make reference to the role of cultural heritage in sustainable development of cities. A reference to these documents in order to include the paper in the actual research debate could be implemented.

From line 202 to 223 different views are recalled (landscape view, architectural view, etc.). They appear as a list of statements that should be supported (for example by bibliographic reference) in order to have more “strength”.

Paragraph 4: the urbanscape emanation concept is clear at theoretical level, but not so much at operational level. The operational consequences of this concept in tools and methods for planning settlements, towns and landscape should be better highlighted.

Line 281: The role of cultural heritage in heritage urbanism should be better underlined in this sentence

Author Response

2.1. The abstract should be integrated in order to clarify the goal of the paper that is not so clear since its beginning. 

ANSWER TO REVIEWER: The supplemented text is – lines: 39-48

Heritage urbanism considers the revitalisation and enhancement of cultural heritage in a spatial, urban and landscape context, and explores models for its inclusion in contemporary life. Heritage is not viewed as isolated objects, but rather as part of the immediate and wider environment. The context/environment affects heritage and its revival, while finding new uses and repurposing heritage has a stimulating effect on the environment and its development. The effects of this interaction can make heritage recognisable and can stimulate its sustainability. The survival and future of heritage are linked to urban and spatial planning which takes into account the integrity of space and the cultural heritage in it. The integral view of cultural heritage requires the concerted action of different fields, such as regional development, the economy, tourism, transportation and infrastructure. A fragmented and selective approach does not yield results. /Rev.2.1./

2.2. The paper is completely lacking in references about the discussed issue although it is a much debated topic at international level. The whole first paragraph should be integrated including bibliographic references (for example in lines 79-80; 121; 129-130; 199). 

ANSWER TO THE REVIEWER: The text has been supplemented with the literature and references have been added to the brackets. Earlier, we only mentioned our work on heritage urbanism to show the way of developing ideas and a more detailed view of heritage urbanism.

Lines 92-105: added references

Lines 155-158: added a red text: A fragmented and selective approach does not yield results, as confirmed by multiple examples of cultural heritage renewal when the renovation is only done according to conservation criteria, without taking into account economic criteria and long-term sustainability.

Lines 166-168: added reference: Ruskin, J., 1981. /Rev 2.2 (line 129-130)/

Lines 259-262: This is an ongoing topic of contemporary discussion. For example, the 2030 Agenda for Sustainable Development (in particular the target 11.4 of the goal 11) and the New Urban Agenda, both by United Nations, make reference to the role of cultural heritage in sustainable development of cities. /Rev. 2.2., 2.9./

All references (old and new)

References

1. Obad Šćitaroci, M., Ed. Cultural Heritage - Possibilities for Spatial and Economic Development / Kulturno naslijeđe - prostorne i razvojne mogućnosti kulturnog naslijeđa, Faculty of Architecture University of Zagreb and Croatian Academy of Sciences and Arts: Zagreb, Croatia, 2015; ISBN 978-953-8042-10-2, ISBN 978-953-8042-11-9 (eBook).

2. Obad Šćitaroci, M., Ed. Modeli revitalizacije i unaprjeđenja kulturnog naslijeđa - multidisciplinarni dijalog, Faculty of Architecture University of Zagreb and Croatian Academy of Sciences and Arts: Zagreb, Croatia, 2017; ISBN 978-953-8042-29-4, ISBN 978-953-8042-30-0 (eBook). 

3. Obad Šćitaroci, M. Heritage as an Active Space and Spatial Resource. In Quality of Life in Urban Landscapes - In Search of a Decision Support System, Cocci Grifoni, R., D’Onofrio, R., Sargolini, R., Eds.; Springer: Cham, Switzerland, 2018, pp. 341– 348, ISBN 978-3-319-65580-2, ISBN 978-3-319-65581-9 (eBook), https://doi.org/10.1007/978-3-319-65581-9

4. Obad Šćitaroci, M.; Bojanić Obad Šćitaroci, B.; Mrđa A., Eds. Cultural Urban Heritage – Development, Learning and Landscape Strategies, Springer: Cham, Switzerland, 2019, ISBN 978-3-030-10611-9, ISBN 978-3-030-10612-6 (eBook), https://doi.org/10.1007/978-3-030-10612-6

5. European Cultural Heritage Strategy for 21th Century, Council of Europe, 2017, Available online:

https://rm.coe.int/16806f6a03 (accessed on 26 April 2019).

6. Smith, L. Uses of Heritage, Routledge (Taylor&Francis Group): London, and New York, 2006, ISBN10:0-415-31830-0, ISBN10:0-203-60226-9 (eBook).

7. Ashworth, G.J.; Graham, B.; Tunbridge, J.E. Pluralising past: heritage, identity and place in multicultural societies, London: Pluto Press: London, 2007.

8. Ashurst, J., Ed. Conservation of Ruins, Elsevier: London, 2006,

ISBN-13: 978-0750664295, ISBN-10: 0750664290.

9. Mendes Zancheti, S.; Simila, K., Measuring Heritage Conservation Performance, VI. International Seminar on Urban Conservation, CECI & ICCROM: Olinda, Brazil; Rome, Italy, 2012, CECI ISBN: 978-85-98747-16-3 ICCROM ISBN: 978-92-9077-230-9.

10. Cohen, N. Urban Planning Conservation and Preservation, McGraw-Hill: New York, 2001,

ISBN-13: 978-0071375849, ISBN-10: 0071375848.

11. Veldpaus,L.; Pereira Roders, A.R.; Colenbrander, B.J.F. Urban Heritage: Putting the Past into the Future, The Historic Environment, 2013, 4(1), 3–18.

12. Janssen, J.; Luiten, E.; Stegmeijer, E. Heritage as sector, factor and vector conceptualizing the shifting relationship between heritage management and spatial planning, European Planning Studies, 2017, 25(9), 1654-1672.

13. McManamon, F.P.; Hatton, A. Eds. Cultural Resource Management in Contemporary Society, Routledge (Taylor&Francis Group): London, 2000, ISBN 978-0-415-64241-5 (ISBN-13: 978-0415117852, ISBN-10: 0415117852).

14. McManamon, F.P. Ed. New Perspectives in Cultural Resource Management, Routledge (Taylor&Francis Group): London, 2017, ISBN 9781138101128.

15. Tunbridge, J.E.; Ashworth, G.J. Dissonant Heritage - the Management of the Past as a Resource in Conflict, John Wiley & Sons: Chichester, UK, 1996, ISBN-13: 978-0471948872, ISBN-10: 047194887X.

16. Baum, M.; Christiaanse, K. Eds. City as Loft - Adaptive Reuse as a Resource for Sustainable Urban Development, GTA Verlag (ETH Zurich): Zurich, Switzerland, 2012, ISBN 978-3-85676-302-2.

17. Recommendation on the Historic Urban Landscape (HUL), UNESCO: Paris, 2011, Available online: https://whc.unesco.org/en/hul/, https://whc.unesco.org/en/news/1026/, https://whc.unesco.org/uploads/activities/documents/activity-638-98.pdf (accessed on 26 April 2019).

18. Ruskin, J. Seven Lamps of Architecture, 8th ed.; Farar, Straus and Giroux: New York, 1981, ISBN: 978-0486261454.

19. Messenger, P.M.; Smith, G.S., Eds. Cultural Heritage Management: A global perspective, University Press of Florida, SAD, 2015, ISBN 978-0-8130-6085-9, ISBN-13: 978-0-8130-34607, DOI: 10.5744/florida/9780813034607.001.0001.

20. Leask A.; Fyall A. Managing world heritage sites. Routledge (Taylor&Francis Group): London and New York, 2006, ISBN-13: 978-0-7506-6546-9, ISBN-10: 0-7506-6546-7. 

21. McKercher, B.; du Cros, H. Cultural Tourism. The Partnership Between Tourism and Cultural Heritage Management, Haworth Hospitality Press (Routledge): London, 2002, ISBN 0‐7890‐1106‐9.

22. Transforming our World: the 2030 Agenda for Sustainable Development

A/RES/70/1, United Nations, Available online: https://sustainabledevelopment.un.org/ (accessed on 26 April 2019).

23. Eppich, R., Ed. Cultural Heritage Landscape and Rural Development, Good Practice, Methodology, Policy Recommendations & Guidelines for Rural Communities, 2014, ISBN: 978-84-697-1389-1.

24. Cultural Landscapes and Herirtage Values, University of Massachusetts Amherst: Amherst, SAD, 2015, Available online: https://www.umass.edu/chs/news/CLHVFinalProgram.pdf (accessed on 26 April 2019).

25. Lardinois, S.; Arato Gonçalves, A. P.; Matarese, L.; Macdonald, S., Eds. Contemporary Architecture in the Historic Environment - An Annotated Bibliography, The Getty Consertvation Institute: Los Angeles, SAD, 2015, Available online:  https://www.getty.edu/conservation/publications_resources/pdf_publications/pdf/cahe_bibliography.pdf (accessed on 26 April 2019).

26. Building in Context - New development in historic areas, English Heritage: London, CABE: London, 2001, Available online: https://www.designcouncil.org.uk/sites/default/files/asset/document/building-in-context-new-development-in-historic-areas.pdf (accessed on 26 April 2019).

27. Butina Watson, G.; Bentley, I. Identity by Design, Butterworth-Heinemann: Oxford, UK, 2007, ISBN-13: 978-0750647670, ISBN-10: 0750647671.

28. Licciardi, G.; Amirtahmasebi, R., Eds. The Economics of Uniqueness – investing in Historic City Cores and Cultural Heritage Assets for Sustainable Development, The World Bank Washington, D.S., 2012, ISBN (paper): 978-0-8213-9650-6ISBN (electronic): 978-0-8213-9706-0, DOI: 10.1596/978-0-8213-9650-6.

29. Navrud, S.; Ready, R.C., Eds. Valuing Cultural Heritage: Applying Environmental Valuation Techniques, Edward Elgar: Cheltenham, UK, 2002, ISBN: 1-84064-079-0.

30. Almo, F. The cultural landscape as a model for the integration of ecology and economiscs, Bioscience, 2000, 50(4), 313-320.

31. Bojanić Obad Šćitaroci, B. Urbanscape Emanation vs. Types of Landscape. In Quality of Life in Urban Landscapes; Cocci Grifoni, R., D’Onofrio, R., Sargolini, R., Eds.; Springer: Cham, Switzerland, 2018, pp. 349–355, https://doi.org/10.1007/978-3-319-65581-9

32. Bojanić Obad Šćitaroci, B. Perceiving Heritage vs. Awareness of Heritage. In Cultural Heritage – Possibilities for Spatial and Economic Development; Mladen Obad Šćitaroci, Ed. Faculty of Architecture University of Zagreb and Croatian Academy of Sciences and Arts: Zagreb, Croatia, 2015, pp. 212-215, ISBN 978-953-8042-10-2, ISBN 978-953-8042-11-9 (eBook).

2.3. At the end of the first paragraph a brief summary about the structure of the paper and the content of the following paragraphs should be included in order to prepare readers for the structure of the paper and contribute to make the goal of the work clearer.

ANSWER TO REVIEWER: added text – lines 120-123

The heritage urbanism approach is presented under the next three titles. This approach aims to enhance cultural heritage, given that it is endangered and is disappearing due to the lack or change of use, or for other reasons. The heritage urbanism approach will be compared to other approaches, both other views on urbanism and approaches focusing on cultural heritage. /Rev.2.3/

2.4. One of the latest documents about the holistic approach necessary in dealing with issues related to cultural heritage is the Historic Urban Landscape Recommendations by UNESCO (2011). This document is never mentioned in the paper. What is the relationship between Heritage Urbanism and UNESCO HUL approach? A section about this relationship should be included, considering the importance and the international interest that the UNESCO approach plays today. 

ANSWERS TO REVIEWER: The text has been completed according to the recommendation

Added literature 1.4 – lines 103-105.

An important starting point of the heritage urbanism approach was the Recommendation on the Historic Urban Landscape (HUL), 2011. [17] This Recommendation focuses mainly on application in the local context. /Rev.2.4/

Added text under the title 3. Heritage Urbanism and other Perspectives – lines 233-239:

The Historic Urban Landscape (HUL) approach is holistic and interdisciplinary. It focuses on historic cities. “It is based on the recognition and identification of a layering and interconnection of natural and cultural, tangible and intangible, international and local values present in any city. It includes social and cultural practices and values, economic processes and the intangible dimensions of heritage as related to diversity and identity” (UNESCO, 2011). [17] Although the starting points are similar, heritage urbanism primarily focuses on potential models for revitalisation and for the enhancement of different types of built heritage, not only historic cities. /Rev.2.4/

2.5. In the paragraph 1.3 there are the research questions. The main research question in this paper should be clearer underlined.

ANSWERS TO REVIEWER: The text has been completed according to the recommendation

Addition: lines 75-78:

The main research question is whether it is possible, based on a number of case studies, to recognise models of the future use of heritage and interpret them as general models that may be applied to numerous specific cases. In doing so, the experience of the past becomes relevant and applicable to contemporary heritage revitalisation and enhancement projects. /Rev.2.5./

2.6. Lines 140-144: Are there some case studies in which this approach has already been implemented?

ANSWERS TO REVIEWER: The text has been completed according to the recommendation

Addition lines 183-188:

The heritage urbanism approach is tested and developed through doctoral research dealing with specific topics and cases. This process continues even after the research has ended, extending to new doctoral research. Practical verification of the approach in several renewal and revitalisation projects involving manor houses and their surroundings is underway. Further application of the approach and its verification have just begun, and this requires some time, so it is not yet possible to speak of the results of practical verification.

For illustrative purposes we list five accepted titles and synopsis of doctoral researches where the heritage urbanism approach is applied. Research works are:

- Integrated Protection Models of Archaeological Heritage in Dubrovnik's historic Area (Zehra Laznibat, Faculty of Architecture University of Zagreb)

- Zagreb Lower Town - Urbanistic Traits of the Eastern Part 1905-2010 (Marijana Sironic, Faculty of Architecture University of Zagreb)

- Urban Traits and Urbanity layer of the King Zvonimir Street in Zagreb (Dario Sironic, Faculty of Architecture University of Zagreb)

- Integrated spatial Protection and Management Models of rural Heritage in protected Areas  - Lonjsko polje Nature Park (Ksenija Petrić, Faculty of Architecture University of Zagreb)

- A Method for Contemporary Architectural Restitution of Croatian Country House Complexes  (Boris Dundović, Vienna University of Technology (TU Wien), Faculty for Architecture and Planning, Institute of History of Art, Building Archaeology and Restoration)

2.7. Lines 146-148: Who are the subjects involved in the identification of the value factors? How are they identified? Are the values identified by experts or also by the support of the community?

ANSWERS TO REVIEWER: The text has been completed according to the recommendation

Addition lines 190-195:

The factors of identity are determined by experts, preferably from different fields and different viewpoints. The identification of these factors is based on historical analyses of various cartographic and written sources, the current situation on the site, and data gathered from literature. The evaluation is made by the experts, but it is also recommended to include the community through questionnaires, workshops and roundtables. /Rev.2.7./

2.8. Lines 152-157: Which are the 17 recognized models? They should be explicitly cited in the paper or bibliographic references need to be included in order to help readers to identify them.

ANSWER TO REVIEWER: The text has been added by specifying references and pages

 – line 206: 

Seventeen models were recognised and divided into three groups: Universal Heritage Models, Basic Heritage Models and Thematic Models of the Heritage Approach. [Obad Šćitaroci, M.; Bojanić Obad Šćitaroci, B.; Mrđa A., Eds., 2019: 457-475] 

It would be too much to include in the text. All types and subtypes are listed in the reference n 4.

1.1. Protection and Conservation Model

1.2. Heritage Revitalisation Model

1.3. Heritage Enhancement Model

1.4. Heritage Re-Use Models

1.5. Economic Heritage Models – Model of Economic Sustainability

1.6. Legal Heritage Model

2. Basic Heritage Models

2.1. Urban/Spatial Heritage Model

2.2. Heritage Transformation Model

2.3. Heritage Integrated Model

2.4. Heritage Interaction Model - multidisciplinary Approach

2.5. Heritage Sustainable Development Model

3. Thematic models of Heritage approach

3.1. System Model

3.2. Architectural and Design Model

3.3. Cultural Turism Model

3.4. Experience Model

3.5. Ambient Authenticity Model

3.6. Landscape-Ecological Model

2.9. Lines 199-200: The contribution of cultural heritage to the development of cities is a great challenge today. In fact, some international debates refer to this topic. For example, the 2030 Agenda (in particular the target 11.4 of the goal 11) and the New Urban Agenda by United Nations make reference to the role of cultural heritage in sustainable development of cities. A reference to these documents in order to include the paper in the actual research debate could be implemented.

ANSWER TO REVIEWER: The text has been completed according to the recommendation

Addition is red - lines 259-262:

The urbanist view studies heritage in the context of the city/settlement through an analysis of how cultural heritage builds the city and how it can contribute to the development of the city/settlement. Its goal is to include heritage into the life of urban and rural areas. This is an ongoing topic of contemporary discussion. For example, the 2030 Agenda for Sustainable Development (in particular target 11.4 of goal 11) and the New Urban Agenda, both by the United Nations, make reference to the role of cultural heritage in the sustainable development of cities. /Rev. 2.2., 2.9/

2.10. From line 202 to 223 different views are recalled (landscape view, architectural view, etc.). They appear as a list of statements that should be supported (for example by bibliographic reference) in order to have more “strength”. 

ANSWERS TO REVIEWER: The text has been completed according to the recommendation

References are added.

The text has been supplemented with the literature and references have been added to the brackets. Earlier, we only mentioned our work on heritage urbanism to show the way of developing ideas and a more detailed view of heritage urbanism.

Added references: lines: 263-286

2.11. Paragraph 4: the urbanscape emanation concept is clear at theoretical level, but not so much at operational level. The operational consequences of this concept in tools and methods for planning settlements, towns and landscape should be better highlighted. 

ANSWERS TO REVIEWER: The text has been completed according to the recommendation

Added text - lines 328-339):

Urbanscape Emanation considers historical, social, cultural, economic, technical and urban research as equal starting points for the interdisciplinary-based spatial planning. This type of research needs cooperation between universities and spatial planning practice which enables easier communication with different stakeholders, experiments in participatory approaches and the constant search for areas to improve both education and planning scenarios and strategies. A multi-layered overlap of a complex network of both people and data is made possible through contemporary technology and GIS-based tools. This overlap reveals both measurable features as well as all invisible, hidden, forgotten and disappeared layers such as social and cultural constructs with psychological influences and with the intentions of raising awareness for the locals and investors. Awareness allows for future towns to draw memory in their landscapes as an added feature and value that can redefine private and public relations leading to better and stronger partnerships. /Rev.2.11./

2.12. Line 281: The role of cultural heritage in heritage urbanism should be better underlined in this sentence.

ANSWERS TO REVIEWER: The text has been completed according to the recommendation

Addition lines 356-359

The main question is not how to prevent the decay of cultural heritage and preserve it. The question is what we can lose if cultural heritage is not included in spaces where people live and in the community to which the heritage belongs. Leaving old heritage to decay while constructing a new one is irrational. /Rev.2.12/

Reviewer 3 Report

Dear Authors,

I have had the opportunity to read your work and I found it very interesting.

In order to make the work more complete, I would have explained how the term "heritage urbanism" was coined and for what purpose (it is only said in lines 25-38 on which occasion it was created and in which works of the authors it was used) and I would have provided a more detailed review of the literature on the subject and on  the related topics (although it is an emerging topic, the bibliography consists only of works by the authors and do not consider the literature of the other topics addressed in the work such as cultural heritage and urban culture).

Author Response

ANSWER TO THE REVIEWER: The text has been supplemented with the literature and references have been added to the brackets. Earlier, we only mentioned our work on heritage urbanism to show the way of developing ideas and a more detailed view of heritage urbanism.

References

1. Obad Šćitaroci, M., Ed. Cultural Heritage - Possibilities for Spatial and Economic Development / Kulturno naslijeđe - prostorne i razvojne mogućnosti kulturnog naslijeđa, Faculty of Architecture University of Zagreb and Croatian Academy of Sciences and Arts: Zagreb, Croatia, 2015; ISBN 978-953-8042-10-2, ISBN 978-953-8042-11-9 (eBook).

2. Obad Šćitaroci, M., Ed. Modeli revitalizacije i unaprjeđenja kulturnog naslijeđa - multidisciplinarni dijalog, Faculty of Architecture University of Zagreb and Croatian Academy of Sciences and Arts: Zagreb, Croatia, 2017; ISBN 978-953-8042-29-4, ISBN 978-953-8042-30-0 (eBook). 

3. Obad Šćitaroci, M. Heritage as an Active Space and Spatial Resource. In Quality of Life in Urban Landscapes - In Search of a Decision Support System, Cocci Grifoni, R., D’Onofrio, R., Sargolini, R., Eds.; Springer: Cham, Switzerland, 2018, pp. 341– 348, ISBN 978-3-319-65580-2, ISBN 978-3-319-65581-9 (eBook), https://doi.org/10.1007/978-3-319-65581-9

4. Obad Šćitaroci, M.; Bojanić Obad Šćitaroci, B.; Mrđa A., Eds. Cultural Urban Heritage – Development, Learning and Landscape Strategies, Springer: Cham, Switzerland, 2019, ISBN 978-3-030-10611-9, ISBN 978-3-030-10612-6 (eBook), https://doi.org/10.1007/978-3-030-10612-6

5. European Cultural Heritage Strategy for 21th Century, Council of Europe, 2017, Available online:

https://rm.coe.int/16806f6a03 (accessed on 26 April 2019).

6. Smith, L. Uses of Heritage, Routledge (Taylor&Francis Group): London, and New York, 2006, ISBN10:0-415-31830-0, ISBN10:0-203-60226-9 (eBook).

7. Ashworth, G.J.; Graham, B.; Tunbridge, J.E. Pluralising past: heritage, identity and place in multicultural societies, London: Pluto Press: London, 2007.

8. Ashurst, J., Ed. Conservation of Ruins, Elsevier: London, 2006,

ISBN-13: 978-0750664295, ISBN-10: 0750664290.

9. Mendes Zancheti, S.; Simila, K., Measuring Heritage Conservation Performance, VI. International Seminar on Urban Conservation, CECI & ICCROM: Olinda, Brazil; Rome, Italy, 2012, CECI ISBN: 978-85-98747-16-3 ICCROM ISBN: 978-92-9077-230-9.

10. Cohen, N. Urban Planning Conservation and Preservation, McGraw-Hill: New York, 2001,

ISBN-13: 978-0071375849, ISBN-10: 0071375848.

11. Veldpaus,L.; Pereira Roders, A.R.; Colenbrander, B.J.F. Urban Heritage: Putting the Past into the Future, The Historic Environment, 2013, 4(1), 3–18.

12. Janssen, J.; Luiten, E.; Stegmeijer, E. Heritage as sector, factor and vector conceptualizing the shifting relationship between heritage management and spatial planning, European Planning Studies, 2017, 25(9), 1654-1672.

13. McManamon, F.P.; Hatton, A. Eds. Cultural Resource Management in Contemporary Society, Routledge (Taylor&Francis Group): London, 2000, ISBN 978-0-415-64241-5 (ISBN-13: 978-0415117852, ISBN-10: 0415117852).

14. McManamon, F.P. Ed. New Perspectives in Cultural Resource Management, Routledge (Taylor&Francis Group): London, 2017, ISBN 9781138101128.

15. Tunbridge, J.E.; Ashworth, G.J. Dissonant Heritage - the Management of the Past as a Resource in Conflict, John Wiley & Sons: Chichester, UK, 1996, ISBN-13: 978-0471948872, ISBN-10: 047194887X.

16. Baum, M.; Christiaanse, K. Eds. City as Loft - Adaptive Reuse as a Resource for Sustainable Urban Development, GTA Verlag (ETH Zurich): Zurich, Switzerland, 2012, ISBN 978-3-85676-302-2.

17. Recommendation on the Historic Urban Landscape (HUL), UNESCO: Paris, 2011, Available online: https://whc.unesco.org/en/hul/, https://whc.unesco.org/en/news/1026/, https://whc.unesco.org/uploads/activities/documents/activity-638-98.pdf (accessed on 26 April 2019).

18. Ruskin, J. Seven Lamps of Architecture, 8th ed.; Farar, Straus and Giroux: New York, 1981, ISBN: 978-0486261454.

19. Messenger, P.M.; Smith, G.S., Eds. Cultural Heritage Management: A global perspective, University Press of Florida, SAD, 2015, ISBN 978-0-8130-6085-9, ISBN-13: 978-0-8130-34607, DOI: 10.5744/florida/9780813034607.001.0001.

20. Leask A.; Fyall A. Managing world heritage sites. Routledge (Taylor&Francis Group): London and New York, 2006, ISBN-13: 978-0-7506-6546-9, ISBN-10: 0-7506-6546-7. 

21. McKercher, B.; du Cros, H. Cultural Tourism. The Partnership Between Tourism and Cultural Heritage Management, Haworth Hospitality Press (Routledge): London, 2002, ISBN 0‐7890‐1106‐9.

22. Transforming our World: the 2030 Agenda for Sustainable Development

A/RES/70/1, United Nations, Available online: https://sustainabledevelopment.un.org/ (accessed on 26 April 2019).

23. Eppich, R., Ed. Cultural Heritage Landscape and Rural Development, Good Practice, Methodology, Policy Recommendations & Guidelines for Rural Communities, 2014, ISBN: 978-84-697-1389-1.

24. Cultural Landscapes and Herirtage Values, University of Massachusetts Amherst: Amherst, SAD, 2015, Available online: https://www.umass.edu/chs/news/CLHVFinalProgram.pdf (accessed on 26 April 2019).

25. Lardinois, S.; Arato Gonçalves, A. P.; Matarese, L.; Macdonald, S., Eds. Contemporary Architecture in the Historic Environment - An Annotated Bibliography, The Getty Consertvation Institute: Los Angeles, SAD, 2015, Available online:  https://www.getty.edu/conservation/publications_resources/pdf_publications/pdf/cahe_bibliography.pdf (accessed on 26 April 2019).

26. Building in Context - New development in historic areas, English Heritage: London, CABE: London, 2001, Available online: https://www.designcouncil.org.uk/sites/default/files/asset/document/building-in-context-new-development-in-historic-areas.pdf (accessed on 26 April 2019).

27. Butina Watson, G.; Bentley, I. Identity by Design, Butterworth-Heinemann: Oxford, UK, 2007, ISBN-13: 978-0750647670, ISBN-10: 0750647671.

28. Licciardi, G.; Amirtahmasebi, R., Eds. The Economics of Uniqueness – investing in Historic City Cores and Cultural Heritage Assets for Sustainable Development, The World Bank Washington, D.S., 2012, ISBN (paper): 978-0-8213-9650-6ISBN (electronic): 978-0-8213-9706-0, DOI: 10.1596/978-0-8213-9650-6.

29. Navrud, S.; Ready, R.C., Eds. Valuing Cultural Heritage: Applying Environmental Valuation Techniques, Edward Elgar: Cheltenham, UK, 2002, ISBN: 1-84064-079-0.

30. Almo, F. The cultural landscape as a model for the integration of ecology and economiscs, Bioscience, 2000, 50(4), 313-320.

31. Bojanić Obad Šćitaroci, B. Urbanscape Emanation vs. Types of Landscape. In Quality of Life in Urban Landscapes; Cocci Grifoni, R., D’Onofrio, R., Sargolini, R., Eds.; Springer: Cham, Switzerland, 2018, pp. 349–355, https://doi.org/10.1007/978-3-319-65581-9

32. Bojanić Obad Šćitaroci, B. Perceiving Heritage vs. Awareness of Heritage. In Cultural Heritage – Possibilities for Spatial and Economic Development; Mladen Obad Šćitaroci, Ed. Faculty of Architecture University of Zagreb and Croatian Academy of Sciences and Arts: Zagreb, Croatia, 2015, pp. 212-215, ISBN 978-953-8042-10-2, ISBN 978-953-8042-11-9 (eBook).

Round  2

Reviewer 2 Report

The paper has been well integrated and it is now much clearer.

Just some comments:

-      Research focus (and thus the goal of the paper), research methods and findings should be more clearly described in the abstract.

-      Line 183-184 - If doctoral theses dealing with these cases have been published, they could be mentioned.

-      References 17, 24, 25, 26 should be aligned with the bibliography format.

-      Reference 17: the third link does not work.

-      Reference 26: the link to the web page does not work.

Author Response

All new corectionn are in green colour

1) Research focus (and thus the goal of the paper), research methods and findings should be more clearly described in the abstract.

Abstract: Heritage urbanism considers the revitalisation and enhancement of cultural heritage in a spatial, urban and landscape context, and explores models for its inclusion in contemporary life. The main research question is whether it is possible, based on a number of case studies, to recognise models of the future use of heritage and interpret them as general models that may be applied to numerous specific cases. In doing so, the experience of the past becomes relevant and applicable to contemporary heritage revitalisation and enhancement projects. The goal of the paper is to present Heritage Urbanism approach as integral view of heritage in line with the ideas of sustainable development. Heritage is not viewed as isolated objects, but rather as part of the immediate and wider environment. The context/environment affects heritage and its revival, while finding new uses and repurposing heritage has a stimulating effect on the environment and its development. The effects of this interaction can make heritage recognisable and can stimulate its sustainability. The survival and future of heritage are linked to urban and spatial planning which takes into account the integrity of space and the cultural heritage in it. Urban and spatial planning methods are used. When these methods are enriched by the heritage urbanism approach, the result is the creation of specific methods that supplement well-known methods. In this context, cultural heritage can be used for place branding, infrastructure development, as a crucial element of urban design, or in other ways that aim to achieve an integral view of cultural heritage. The integral view requires the concerted action of different fields, such as regional development, the economy, tourism, transportation and infrastructure. A fragmented and selective approach does not yield results.

2) Line 183-184 - If doctoral theses dealing with these cases have been published, they could be mentioned.

PhD studies are ongoing, not yet completed and not published. The synopsis of these doctoral studies has been defended. So we do not cite them.

3) References 17, 24, 25, 26 should be aligned with the bibliography format.

17. Recommendation on the Historic Urban Landscape. Available online: https://whc.unesco.org/en/hul/, https://whc.unesco.org/en/news/1026/, https://whc.unesco.org/uploads/activities/documents/activity-638-98.pdf (accessed on 4 May 2019).

24. Cultural Landscapes and Herirtage Values. Available online: https://www.umass.edu/chs/news/CLHVFinalProgram.pdf (accessed on 4 May 2019).

25. Lardinois, S.; Arato Gonçalves, A. P.; Matarese, L.; Macdonald, S., Eds. Contemporary Architecture in the Historic Environment - An Annotated Bibliography. Available online:  https://www.getty.edu/conservation/publications_resources/pdf_publications/pdf/cahe_bibliography.pdf (accessed on 4 May 2019).

26. Building in Context - New development in historic areas. Available online: https://www.designcouncil.org.uk/sites/default/files/asset/document/building-in-context-new-development-in-historic-areas.pdf (accessed on 4 May 2019).

4) Reference 17: the third link does not work.

We can open

5) Reference 26: the link to the web page does not work.

We can open

https://www.designcouncil.org.uk/sites/default/files/asset/document/building-in-context-new-development-in-historic-areas.pdf